# Is obesity more likely among children sharing a household with an older child with obesity? Cross-sectional study of linked National Child Measurement Programme data and electronic health records

Nicola Firman [1], Marta Wilk,[1] Milena Marszalek,[1] Lucy Griffiths [2], Gill Harper,[1] Carol Dezateux[1]

Society for Social Medicine and Population Health annual scientific meeting 2023 and the UK Congress on Obesity 2023

¹Wolfson Institute of Population Health, Queen Mary University of London, London, UK
²Population Data Science, Swansea University Medical School, Swansea, UK

**Correspondence to**
Nicola Firman; nicola.firman@qmul.ac.uk

## ABSTRACT

**Background/objectives** We identified household members from electronic health records linked to National Child Measurement Programme (NCMP) data to estimate the likelihood of obesity among children living with an older child with obesity.

**Methods** We included 126 829 NCMP participants in four London boroughs and assigned households from encrypted Unique Property Reference Numbers for 115 466 (91.0%). We categorised the ethnic-adjusted body mass index of the youngest and oldest household children (underweight/healthy weight <91st, ≥91st overweight <98th, obesity ≥98th centile) and estimated adjusted ORs and 95% CIs of obesity in the youngest child by the oldest child's weight status, adjusting for number of household children (2, 3 or ≥4), youngest child's sex, ethnicity and school year of NCMP participation.

**Results** We identified 19 702 households shared by two or more NCMP participants (% male; median age, range (years)—youngest children: 51.2%; 5.2, 4.1–11.8; oldest children: 50.6%; 10.6, 4.1–11.8). One-third of youngest children with obesity shared a household with another child with obesity (33.2%; 95% CI: 31.2, 35.2), compared with 9.2% (8.8, 9.7) of youngest children with a healthy weight. Youngest children living with an older child considered overweight (OR: 2.33; 95% CI: 2.06, 2.64) or obese (4.59; 4.10, 5.14) were more likely to be living with obesity.

**Conclusions** Identifying children sharing households by linking primary care and school records provides novel insights into the shared weight status of children sharing a household. Qualitative research is needed to understand how food practices vary by household characteristics to increase understanding of how the home environment influences childhood obesity.

## INTRODUCTION

Childhood obesity is a major public health concern globally and reflects a complex number of factors, in particular socioeconomic inequalities.[1] In England, more than one-quarter of children leave primary school with overweight or obesity at a level of severity defined as in need of clinical intervention.[2]

A child's health, including their weight status, is significantly affected by the environment in which they live. Better understanding of households, their composition, and the health of children and adults who share households may provide novel actionable insights to address unhealthy weight in childhood.

Research has shown that child obesity is associated with parental obesity, where parental overweight or obesity is associated

---

### WHAT IS ALREADY KNOWN ON THIS TOPIC

⇒ There is evidence to suggest that children living with older siblings with obesity are more likely to be living with obesity themselves. Research to date has largely focused on the weight status of biological siblings.

### WHAT THIS STUDY ADDS

⇒ We examined associations between child household weight status using novel linkages between school measurement and electronic health records. We showed that younger children living with an older child with obesity were more than four times more likely to live with obesity than those living with an older child with a healthy weight.

### HOW THIS STUDY MIGHT AFFECT RESEARCH, PRACTICE OR POLICY

⇒ Household factors are potentially more modifiable than genetic or prenatal influences. Taking a household-level approach could potentially reach more children living with, and at risk of, obesity.

with an increased risk of obesity in their child.[3] This relationship is stronger for mothers than fathers.[4 5] Less is known about the associations between the obesity status of child household members. Research investigating associations between siblings' weight status has reported inconsistent results.[6–13] Children living together may experience similar genetic, environmental and socioeconomic circumstances, which may in turn contribute to a shared risk of obesity. The shared household environment is potentially more modifiable than genetic or prenatal influences.

A 2023 systematic review identified that siblings' health-related behaviours and weight-related outcomes varied according to sibling sex and birth order.[14] Our understanding of how household composition, including presence of a sibling or other household children, as well as their weight status, influences childhood obesity could be improved with further research which includes all child household members and not just those who are biologically related.

We identified individuals sharing a household using electronic health records and linked this to school measurement programme data to estimate the likelihood of obesity among children living with an older child with obesity. We hypothesised that younger children will be more likely to be living with obesity if they share a household with an older child living with obesity. We also investigated whether household composition and size, and dwelling type, influenced the likelihood of childhood obesity.

## METHODS
### Study population
Children in the first (reception year) and last (year 6) years of primary school are invited to participate in the National Child Measurement Programme (henceforth known as the school measurement programme), which measures the height and weight of children aged 4–5 and 10–11 years old attending state-maintained schools in England. More than 1 million children take part annually, with participation rates remaining higher than 90% since 2010/2011.[15] School participation is voluntary, although over 99% participate.[16] In City & Hackney, approximately one-quarter of school-aged children attend private or faith schools, compared with equivalent figures of 1.4%, 5.0% and 5.0% for Newham, Tower Hamlets and Waltham Forest, respectively.[17] We do not have information about the small proportion of children who opt out of the school measurement programme.

We linked 126 829 of 128 544 (98.7%) school measurement programme records from four northeast London local authorities (City & Hackney, Newham, Tower Hamlets and Waltham Forest) to general practice (GP) electronic health records via the Discovery Data Service.[18]

### Data sources
We obtained pseudonymised school measurement programme data for the 2013/2014–2018/2019 academic years under data processing agreements with each local authority public health department. We only received school measurement programme records that had been returned to each local authority after quality assurance checks.[19] As the available date of school measurement programme measurement was restricted to month and year, we randomly assigned a day of measurement within term time, excluding weekends and bank holidays to avoid a spurious reduction in variance in age at measurement occasioned by using the same fixed date of measurement for every child (R Studio; V.1.0.153; code available here: bit.ly/random_day).

Pseudonymised data were provided from the Discovery Data Service which receives primary care electronic health records on a daily basis from all GPs in northeast London. Demographic and clinical data recorded up to 1 November 2021 were extracted for school measurement programme participants successfully linked to the Discovery Data Service via pseudonymised National Health Service (NHS) numbers created using Open-Pseudonymiser software.[20] All data were extracted and managed according to UK NHS information governance requirements.[21]

### Data processing
#### Residential Anonymised Linkage Fields
Every addressable location in Great Britain is assigned a Unique Property Reference Number (UPRN). UPRNs identify a place of residence at a granular level, identifying individual properties, for example, houses or flats within a block or building shell. UPRNs are allocated to GP-recorded addresses using the validated ASSIGN algorithm,[22] and pseudonymised into Residential Anonymised Linkage Fields (RALFs) within the Discovery Data Service, using a study-specific encryption key.

#### Identifying household members at the child's school measurement date
A household can only be defined at a single point in time because people living at an address may change over time while the UPRN assigned to the residential dwelling stays the same.

A data extract containing all RALFs associated with any address(es) recorded in a child's electronic health record was extracted. The file contained start and end dates of patient registration (enrolment) with the GP as well as address start and end dates. Address start and end dates refer to the dates at which a patient lived at a particular address. In most cases, these align with GP registration dates, but could differ, if for example, a patient moved house but remained registered with the same GP.

Figure 1 describes the process for deciding which, if any, of the child's RALF was the place of residence at the time of their school measurement programme measurement. If the school measurement programme date of

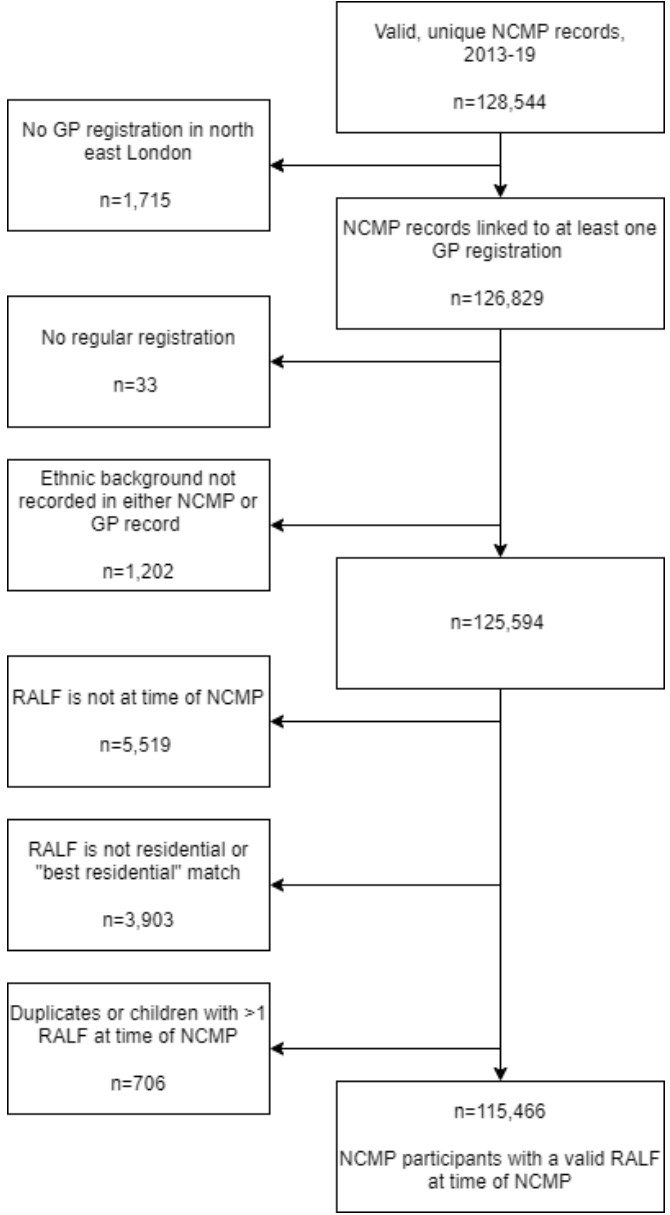

**Figure 1** Identifying a valid Residential Anonymised Linkage Field (RALF) at the time of National Child Measurement Programme (NCMP) measurement. Children living in non-residential dwellings or where the UPRN match qualifier was not a 'best' residential match were excluded (n=3903). The match qualifier indicates how close to the place of residence the assigned UPRN is. We excluded 3903 without residential RALF or best match RALF qualifier. In most cases (66.8%), the RALF assigned to these GP-recorded addresses was associated with a property shell, rather than the exact individual property. Others were living in sheltered accommodation or houses of multiple occupancy not further divided to enable household identification. The prevalence of overweight and obesity among the excluded children was similar to estimates among index children. GP, general practice; UPRN, Unique Property Reference Number.

measurement took place between the most recent of the registration and address start dates and the earliest of the registration and address end dates, the RALF was considered to be the place of residence at the time of school measurement programme (online supplemental figure 1). Children without RALF at the time of their school measurement programme were excluded (n=5519). We retained 115 466 children with RALF at the time of their school measurement programme measurement (referred to hereafter as index children).

### Identifying other National Child Measurement Programme participants in the household

Similar steps were taken to identify other school measurement programme participants sharing a household with index children. We started with 122 759 school measurement programme participants with at least one RALF (online supplemental figure 2) and included in 'dataset two'.

Child pairs were excluded if the index child's school measurement programme measurement date did not fall within dataset two child's RALF dates (online supplemental figure 3). Pairs were also excluded if the dataset two child was younger than the index child or if the dataset two child's school measurement programme measurement was after the index child's. This process found the youngest index child and identified the oldest school measurement programme participant living in the same household at the time of the index child's school measurement programme measurement. Of 128 554 school measurement programme participants, 21 623 youngest/oldest child pairs were identified.

### Identifying other household members and deriving household characteristics

We identified all people who had ever lived at any of the RALFs identified in the dataset of youngest/oldest child pairs. Steps were taken to determine household members at the time of the index child's school measurement programme measurement (online supplemental figure 4). Full household information was identified for 19 702 youngest/oldest child pairs.

### Outcome of interest

Obesity status of the index child was defined by the UK1990 clinical reference standard.[23] After application of ethnic-specific body mass index (BMI) adjustments,[24] a binary variable indicating obesity was defined as a BMI greater than or equal to the 98th age-specific and sex-specific centile. The index child's ethnic-adjusted BMI z-score was a secondary outcome.

### Explanatory variables

Ethnic-specific BMI adjustments[24] were applied to the older child's BMI, and weight status determined according to the UK1990 clinical reference standard[23] categorising BMI into one of four mutually exclusive groups: 'underweight' (<2nd centile), 'healthy weight' (≥2nd–<91st centile), 'overweight' (≥91st–<98th centile) or 'obese' (≥98th centile). The older child's BMI z-score was also considered as an explanatory variable.

**Table 1** Sample characteristics of index children participating in the National Child Measurement Programme (NCMP)

| | All (n=19702) | | | Reception (n=13699) | | | Year 6 (n=6003) | | |
|---|---|---|---|---|---|---|---|---|---|
| | n | % | 95% CI | n | % | 95% CI | n | % | 95% CI |
| **Sex** | | | | | | | | | |
| Male | 10079 | 51.2 | 50.5, 51.9 | 7005 | 51.2 | 50.4, 52.1 | 3074 | 51.1 | 49.9, 52.4 |
| Female | 9623 | 48.8 | 48.1, 49.5 | 6694 | 48.8 | 47.9, 49.6 | 2929 | 48.9 | 47.6, 50.1 |
| **School year*** | | | | | | | | | |
| Reception | 13699 | 69.5 | 68.9, 70.2 | 13699 | 100.0 | | | | |
| Year 6 | 6003 | 30.5 | 29.8, 31.1 | | | | 6003 | 100.0 | |
| **Academic year†** | | | | | | | | | |
| 2031/2014 & 2014/2015 | 517 | 2.6 | 2.4, 2.9 | 463 | 3.4 | 3.1, 3.7 | 54 | 0.9 | 0.6, 1.2 |
| 2015/2016 | 1926 | 9.8 | 9.4, 10.2 | 1530 | 11.2 | 10.7, 11.7 | 396 | 6.6 | 6.0, 7.3 |
| 2016/2017 | 3751 | 19.0 | 18.5, 19.6 | 2748 | 20.0 | 19.4, 20.7 | 1003 | 16.7 | 15.8, 17.7 |
| 2017/2018 | 5980 | 30.4 | 29.7, 31.0 | 4100 | 30.0 | 29.2, 30.7 | 1880 | 31.3 | 30.1, 32.5 |
| 2018/2019 | 7528 | 38.2 | 37.5, 38.9 | 4858 | 35.5 | 34.7, 36.3 | 2670 | 44.5 | 43.2, 45.7 |
| **Local authority‡** | | | | | | | | | |
| City & Hackney | 4998 | 25.4 | 24.8, 26.0 | 3489 | 25.5 | 24.7, 26.2 | 1509 | 25.2 | 24.1, 26.3 |
| Newham | 6472 | 32.9 | 32.2, 33.5 | 4444 | 32.5 | 31.7, 33.3 | 2028 | 33.8 | 32.6, 35.0 |
| Tower Hamlets | 3495 | 17.7 | 17.2, 18.3 | 2571 | 18.7 | 18.1, 19.4 | 924 | 15.4 | 14.5, 16.3 |
| Waltham Forest | 4737 | 24.0 | 23.4, 24.6 | 3195 | 23.3 | 22.6, 24.0 | 1542 | 25.6 | 24.6, 26.8 |
| **Ethnic background§** | | | | | | | | | |
| White | 4615 | 23.4 | 22.9, 24.0 | 3240 | 23.7 | 22.9, 24.4 | 1375 | 22.9 | 21.9, 24.0 |
| Mixed and other | 3823 | 19.4 | 18.8, 19.9 | 2620 | 19.1 | 18.4, 19.8 | 1203 | 20.0 | 19.0, 21.1 |
| South Asian | 6812 | 34.6 | 33.9, 35.3 | 4813 | 35.1 | 34.3, 35.9 | 1999 | 33.3 | 32.1, 34.5 |
| Black | 4452 | 22.6 | 22.0, 23.2 | 3026 | 22.1 | 21.4, 22.8 | 1426 | 23.7 | 22.7, 24.8 |
| **Weight status¶** | | | | | | | | | |
| Underweight | 270 | 1.4 | 1.2, 1.5 | 194 | 1.4 | 1.2, 1.6 | 76 | 1.3 | 1.0, 1.6 |
| Healthy weight | 15005 | 76.2 | 75.6, 76.8 | 11025 | 80.5 | 79.9, 81.2 | 3980 | 66.3 | 65.1, 67.5 |
| Overweight | 2372 | 12.0 | 11.5, 12.4 | 1399 | 10.2 | 9.7, 10.7 | 973 | 16.1 | 15.2, 17.0 |
| Obese | 2055 | 10.4 | 10.0, 10.9 | 1081 | 7.9 | 7.4, 8.3 | 974 | 16.3 | 15.4, 17.3 |

*School year of participation in the NCMP; reception participants are aged 4–5 years and year 6 participants are aged 10–11 years.
†Academic year of participation in the NCMP. Academic years run from September to July. The 2013/2014 and 2014/2015 academic years were combined to minimise the risk of disclosing individuals.
‡Local authority of school where child participated in the NCMP.
§As recorded in the NCMP and, where missing, supplemented with ethnic background as recorded in the child's primary care electronic health record.
¶NCMP-recorded body mass index (BMI) after application of ethnic-specific BMI adjustments, categorised according to UK1990 clinical reference standard: 'underweight' (BMI <2nd centile), 'healthy weight' (≥2nd–<91st centile), 'overweight' (≥91st–<98th centile) or 'obese' (≥98th centile).

School measurement programme-recorded sex, local authority of the school where the child participated in the school measurement programme, academic year (September–July) and school year (reception/year 6) of participation in the school measurement programme were explanatory variables.

School measurement programme-recorded ethnic background was grouped into four mutually exclusive groups[25]: white ('white British', 'white Irish' or 'any other white background'); black ('black African', 'black Caribbean' or 'any other black background'); South Asian ('Indian', 'Pakistani', 'Bangladeshi' or 'Sri Lankan'); and a combination of mixed and other ('any other ethnic background', 'mixed ethnicity', 'Chinese' or 'Asian other'). Where ethnic background was missing or not stated in the school measurement programme, ethnic background as recorded in the electronic health record (n=11077) was used.

An area-level measure of relative deprivation—Index of Multiple Deprivation (IMD) decile[26]—was assigned to each school measurement programme record based on the postcode of the child's home address. IMD decile

**Table 2** Household characteristics of children living in households with two National Child Measurement Programme participants

| | Two school measurement programme-participant households (n=19 702) | | |
|---|---|---|---|
| | n | % | 95% CI |
| IMD quintile* | | | |
| 1 (most deprived) | 10 375 | 52.6 | 51.9, 53.3 |
| 2 | 7836 | 39.8 | 39.1, 40.5 |
| 3 | 1292 | 6.6 | 6.2, 6.9 |
| 4 | 156 | 0.8 | 0.7, 0.9 |
| 5 (least deprived) | 43 | 0.2 | 0.2, 0.3 |
| Number of children in the household | | | |
| 2 | 6449 | 32.8 | 32.1, 33.4 |
| 3 | 7228 | 36.6 | 36.0, 37.3 |
| 4 or more | 6025 | 30.6 | 30.0, 31.2 |
| Household composition | | | |
| Working adults with children | 14 976 | 76.0 | 75.4, 76.6 |
| Single working-aged adult with children | 2873 | 14.6 | 14.1, 15.1 |
| Three generation and skipped generation | 1853 | 9.4 | 9.0, 9.8 |
| Property classification | | | |
| Flat | 10 260 | 52.1 | 51.4, 52.8 |
| Terraced house | 8154 | 41.4 | 40.7, 42.1 |
| Other | 1288 | 6.5 | 6.2, 6.9 |

*2015 Index of Multiple Deprivation (IMD) quintile assigned based on the child's home address postcode as recorded by the school where the child participated in the National Child Measurement Programme. The 2015 IMD accounts for socioeconomic characteristics in lower layer super output areas (LSOAs), small geographies typically comprising an average population of 1500 people or 650 households. IMD score is derived from Indices of Deprivation, which cover seven domains: income; employment; education, skills and training; health; crime; barriers to housing and services; and living environment. The IMD score for each LSOA in England is ranked, from most to least deprived, and divided into 10 equal groups indicating the most deprived 10% of LSOAs to the least deprived 10% of LSOAs, nationally. The school measurement programme dataset includes each child's IMD 2015 score and decile.
IMD, Index of Multiple Deprivation.

was concatenated into five quintiles ranging from most to least deprived.

A categorical variable was derived from a count of children (aged <18.0 years) assigned the same RALF as the school measurement programme participant, grouped as follows: 2; 3–4; 5–6; 7–10.

We categorised household composition using a modified Harper and Mayhew method[27] by counting the number of household members in three age brackets: 0–17 years (children), 18–64 years (working-aged adults) and 65 or older (older adults) and grouping into: working-aged adults with children; a single working-aged adult with children; at least one working-aged and one older adult with children (three-generation household), or at least one older adult with children (skipped-generation household).

The property classification, as given by the ASSIGN algorithm, categorised properties into three groups: flats, terraced houses and other.

Sex concordance was coded either the same (when both children shared the same sex) or different (when the two children had differing sexes). The time difference between the youngest and oldest children's school measurement programme measurements was calculated as the time in months between the two measurements.

### Statistical analyses
We estimated the prevalence of obesity among children living with an older school measurement programme participant and explored variation by the weight status of the older child. We used binary logistic regression to estimate the likelihood of obesity in the index child (OR and 95% CI) by the older child's weight status, after accounting for individual and household characteristics. We conducted linear regression to estimate the effect of a one-unit increase in the oldest child's BMI $z$-score on the index child's BMI $z$-score, after checking residuals were normally distributed. All analyses, conducted using Stata (MP/V.15.0), were stratified by school year.

### Patient and public involvement
This research was done without patient or public involvement. Neither were invited to comment on the study design and were not consulted to develop relevant outcomes or interpret results.

### RESULTS
Index children were, by study design, more likely to take part in the school measurement programme in the reception school year and in the more recent academic years (table 1). Similarly, the oldest children were more likely to take part in the school measurement programme in year 6 and in the earlier academic years (online supplemental table 1). 7.9% of reception year youngest children and 16.3% of year 6 youngest children were living with obesity (table 1). Equivalent estimates using International Obesity Task Force cut-offs are reported in online supplemental table 2. Three-quarters lived in households with adults of working age only, and more than half lived in flats (table 2).

A greater proportion of index children with obesity were male, participating in the school measurement programme in year 6 and in Tower Hamlets and Newham, and from South Asian ethnic backgrounds, compared with index children with underweight or a healthy weight (table 3).

**Table 3** Sociodemographic characteristics of index children living in households with two National Child Measurement Programme (NCMP) participants, by index child's weight status*

| | Underweight & healthy weight (n=15275) | | | Overweight (n=2372) | | | Obese (n=2055) | | |
|---|---|---|---|---|---|---|---|---|---|
| | n | % | 95% CI | n | % | 95% CI | n | % | 95% CI |
| **Sex** | | | | | | | | | |
| Male | 7636 | 50.0 | 49.3, 50.8 | 1256 | 52.9 | 50.9, 54.9 | 1187 | 57.8 | 55.6, 59.8 |
| Female | 7639 | 50.0 | 49.2, 50.7 | 1116 | 47.1 | 45.1, 49.1 | 868 | 42.2 | 40.2, 44.4 |
| **School year†** | | | | | | | | | |
| Reception | 11 219 | 73.4 | 72.7, 74.1 | 1399 | 59.0 | 57.0, 61.0 | 1081 | 52.6 | 50.3, 54.6 |
| Year 6 | 4056 | 26.6 | 25.9, 27.3 | 973 | 41.0 | 39.0, 43.0 | 974 | 47.4 | 45.4, 49.7 |
| **Academic year‡** | | | | | | | | | |
| 2013/2014 & 2014/2015 | 421 | 2.8 | 2.5, 3.0 | 49 | 2.1 | 1.6, 2.7 | 47 | 2.2 | 1.7, 3.0 |
| 2015/2016 | 1506 | 9.9 | 9.4, 10.3 | 219 | 9.3 | 8.2, 10.5 | 201 | 9.8 | 8.6, 11.2 |
| 2016/2017 | 2938 | 19.2 | 18.6, 19.9 | 434 | 18.3 | 16.8, 19.9 | 379 | 18.3 | 16.7, 20.0 |
| 2017/2018 | 4639 | 30.4 | 29.7, 31.1 | 741 | 31.1 | 29.3, 33.0 | 600 | 29.2 | 27.2, 31.1 |
| 2018/2019 | 5771 | 37.7 | 37.0, 38.5 | 929 | 39.2 | 37.3, 41.2 | 828 | 40.5 | 38.3, 42.6 |
| **Local authority§** | | | | | | | | | |
| City & Hackney | 4001 | 26.2 | 25.5, 26.9 | 556 | 23.2 | 21.5, 24.9 | 441 | 21.5 | 19.8, 23.3 |
| Newham | 4905 | 32.2 | 31.4, 32.9 | 822 | 34.6 | 32.7, 36.5 | 745 | 36.3 | 34.3, 38.5 |
| Tower Hamlets | 2572 | 16.8 | 16.2, 17.4 | 468 | 19.9 | 18.4, 21.6 | 455 | 22.1 | 20.3, 23.9 |
| Waltham Forest | 3797 | 24.8 | 24.2, 25.5 | 526 | 22.3 | 20.7, 24.0 | 414 | 20.1 | 18.4, 21.9 |
| **Ethnic background¶** | | | | | | | | | |
| White | 3739 | 24.5 | 23.8, 25.2 | 522 | 22.0 | 20.4, 23.7 | 354 | 17.2 | 15.6, 18.9 |
| Mixed and other | 3052 | 20.0 | 19.3, 20.6 | 412 | 17.4 | 15.8, 18.9 | 359 | 17.5 | 15.9, 19.2 |
| South Asian | 4677 | 30.6 | 29.9, 31.3 | 1082 | 45.6 | 43.8, 47.8 | 1053 | 51.2 | 49.1, 53.4 |
| Black | 3807 | 24.9 | 24.3, 25.6 | 356 | 15.0 | 13.5, 16.4 | 289 | 14.1 | 12.6, 15.6 |

*NCMP-recorded body mass index (BMI) after application of ethnic-specific BMI adjustments, categorised according to UK1990 clinical reference standard: 'underweight or healthy weight' (<91st centile), 'overweight' (≥91st–<98th centile) or 'obese' (≥98th centile).
†School year of participation in the NCMP; reception participants are aged 4–5 years and year 6 participants are aged 10–11 years.
‡Academic year of participation in the NCMP. Academic years run from September to July. The 2013/2014 and 2014/2015 academic years were combined to minimise the risk of disclosing individuals.
§Local authority of school where child participated in the NCMP.
¶As recorded in the NCMP and, where missing, supplemented with ethnic background as recorded in the child's primary care electronic health record.

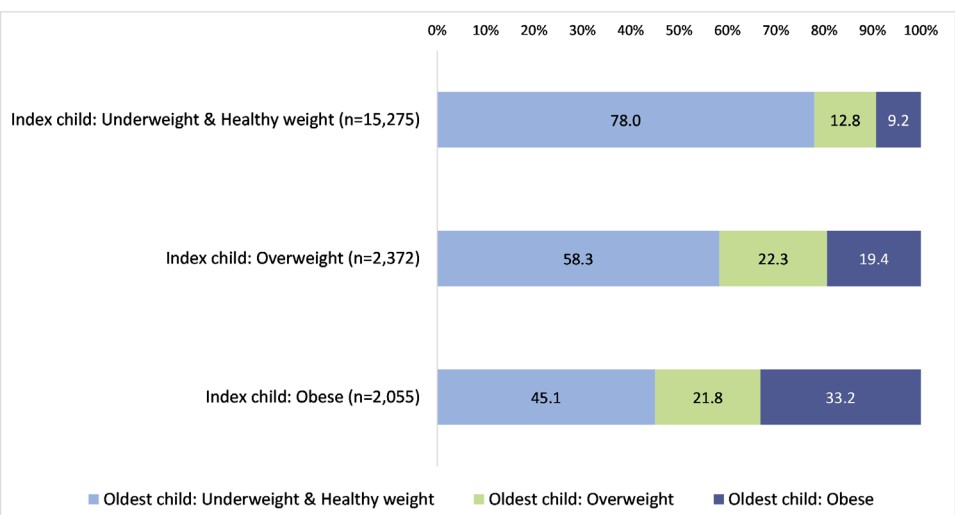

**Figure 2** Oldest child's weight status stratified by index child's weight status[1]. [1]National Child Measurement Programme-recorded body mass index (BMI) after application of ethnic-specific BMI adjustments, categorised according to UK1990 clinical reference standard: 'underweight or healthy weight' (<91st centile), 'overweight' (≥91st–<98th centile) or 'obese' (≥98th centile).

There was no variation in the number of children sharing a household or property classification by weight status (online supplemental table 3). A lower proportion of index children living with obesity lived in households with a single adult (12.5%; 95% CI: 11.2, 14.0) compared with the proportion among children with an underweight/healthy weight status (14.9%; 14.4, 15.5).

One-fifth and one-third of index children living with obesity shared a household with another child with overweight or obesity, respectively, higher than those with underweight or of a healthy weight (online supplemental table 4 and figure 2). Sex concordance, nor time difference between the index and older children's school measurement programme measurement dates, did not vary by weight status of the index child.

In adjusted analyses, index children living with an older child with overweight or obesity were more likely to be living with obesity. Conversely, those sharing a household with two other children were less likely to be living with obesity (figure 3; univariable and multivariable results are presented in online supplemental table 5).

In multivariable linear regression, a one-unit increase in the oldest child's BMI $z$-score was associated with a 0.32 (95% CI: 0.30, 0.33) increase in the index child's BMI $z$-score. Similarly, in linear regression models stratified by the school year of participation in the school measurement programme, a one-unit increase in the oldest child's BMI $z$-score predicted a 0.28 (0.27, 0.29) and 0.38 (0.35, 0.40) increase in reception and year 6 index child's BMI $z$-scores, respectively.

## DISCUSSION
### Summary of key findings
We examined associations between child household weight status using novel linkages between school measurement and electronic health records. We showed that younger children living with an older child with obesity were more than four times more likely to live with obesity than those living with an older child with a healthy weight.

### Strengths and limitations
We used UK1990 clinical thresholds to identify children with obesity considered in need of clinical intervention, as advised by the Scientific Advisory Committee on Nutrition,[23] in an ethnically diverse area of London with high levels of childhood obesity. We recognise these cut-offs do not allow for international comparisons. Our findings may not be generalisable to areas in the UK with lower levels of deprivation and ethnic diversity. The school measurement programme has high participation rates, but our study sample did not include children attending non-state-maintained schools of which there is a higher proportion in City & Hackney.

We used linked school measurement programme records of weight status as we have previously shown that GP electronic health records do not contain accurate, up-to-date child measurement data and are biased to children at both extremes of the BMI distribution.[28] This resulted in exclusion of 30 552 school measurement programme participants who did not live with another school measurement programme participant in the 2013–2019 academic years.

We used a robust methodology to identify household members at the time of the school measurement programme measurement. The ASSIGN algorithm has been shown to match 98.6% of primary care patient addresses to UPRNs.[22] We adopted a conservative approach to identify 'true' household members, by excluding school measurement programme participants living in large or non-residential households. It is possible that we included patients who no longer live at their registered address (so would not consult with their registered GP). There is also likely to be a time lag between a

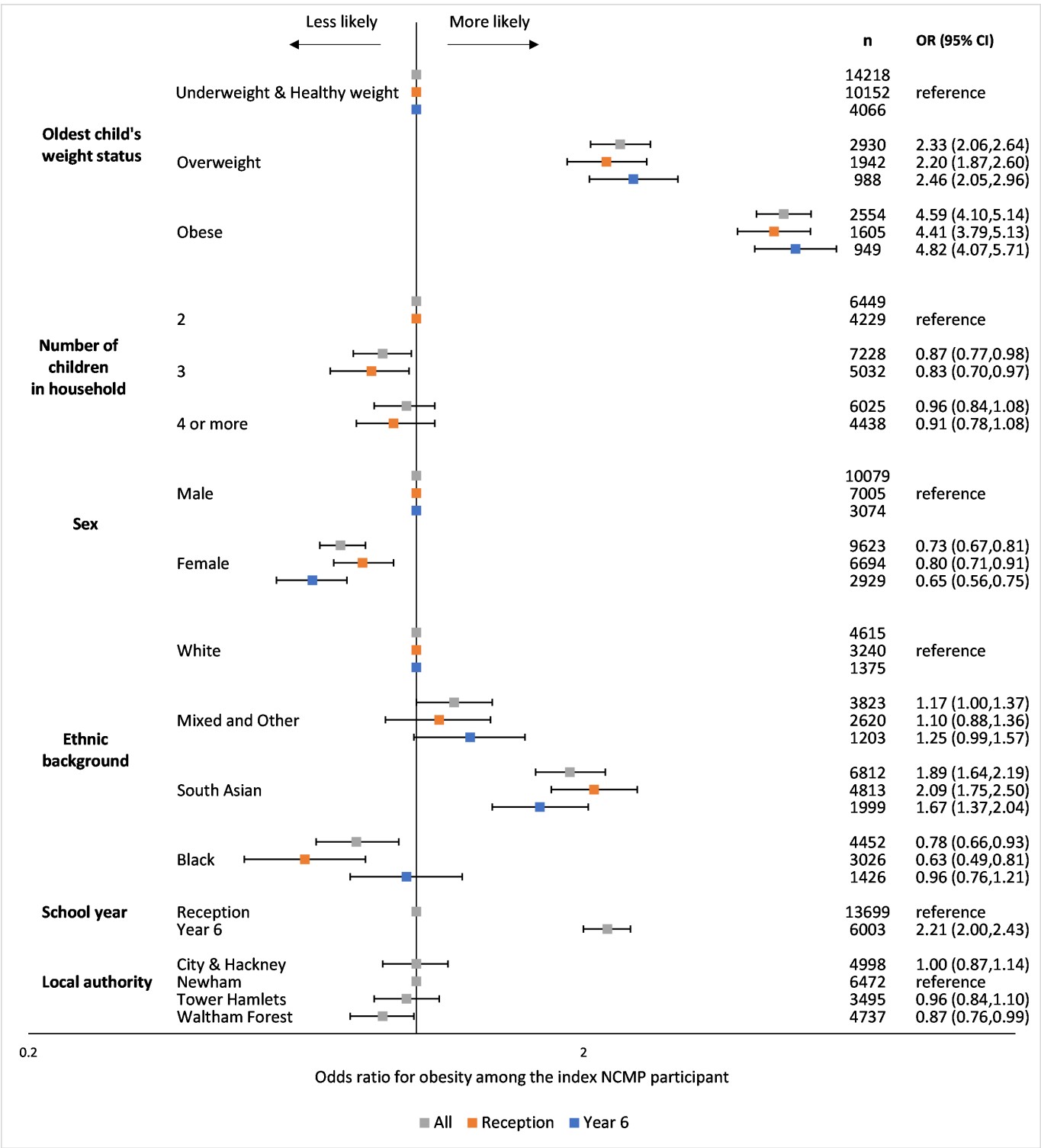

**Figure 3** Multivariable[1] OR estimating the likelihood of obesity[2] among the youngest children living in households with two National Child Measurement Programme (NCMP) participants. [1]The model including all households with two NCMP participants mutually adjusted for oldest child's weight status, number of children in the household, sex, ethnic background as recorded in the NCMP and, where missing, supplemented with ethnic background as recorded in the child's primary care electronic health record, school year of participation in the NCMP (reception participants are aged 4–5 years and year 6 participants are aged 10–11 years) and local authority of school where child participated in the NCMP. The model which only included households where the youngest child participated in the NCMP in reception year mutually adjusted for the oldest child's weight status, number of children in the household, sex and ethnic background. The model which only included households where the youngest child participated in the NCMP in year 6 mutually adjusted for the oldest child's weight status, sex and ethnic background. [2]NCMP-recorded body mass index (BMI) after application of ethnic-specific BMI adjustments, categorised according to UK1990 clinical reference standard: 'obese' (≥98th centile). ORs are plotted on a logarithmic scale.

patient's GP registrations, and a period of time where a patient has moved from an area but remains registered with a GP. Hence, we may have overestimated the true number of household members.

We were not able to determine whether child household members were biologically related. Similarly, we were not able to identify biological parents and account for parental BMI in our analyses.

## Comparison with existing literature

Our findings support those reporting an increased likelihood of obesity among children living with other children with obesity.[12 14] There are likely to be several explanations for this. First, children in the same household spend their time together and share the same resources, which supports the notion of the 'shared home environment'.[29] Siblings eat similar diets, and participate in similar levels of physical activity and sedentary behaviours.[14] Others note that older children are important influencers in children's health-related behaviours, particularly healthy eating. Younger children want to copy the behaviours of their older siblings, explaining the positive correlation between both children's healthy and unhealthy behaviours.[30 31] Children living in the same household are likely to be exposed to the same level of family income, and potentially the same diet and physical activity.[32] Outside of the home, children will be exposed to the same built environment. Finally, biologically related children sharing the same household may share a common genetic predisposition to obesity.[33]

## Implications for research, policy and practice

Our findings highlight the importance of understanding the household distribution of childhood obesity when designing services in populations with high prevalences of obesity and limited resources. A household-level approach may be a pragmatic response to identifying higher-risk households by considering information about all resident children. The shared household environment is potentially more modifiable than genetic or prenatal influences, and analyses of the shared weight status of household members provide new insights into people sharing the same living space, regardless of their biological relationships. This insight is particularly pertinent now that children are increasingly living with household members with whom they may have no biological relationship.[34]

Routinely collected electronic health records provide a limited view of the home environment, and further qualitative research is necessary to fully understand who the decision-makers are, and how practices and attitudes relating to food purchasing and diet, as well as physical activity opportunities, are negotiated on a daily basis.

## Conclusion

Children living with an older child with obesity are more likely to be living with obesity. Early intervention should be approached from a household perspective which takes into account the roles of, and implications for, all household members.

**Acknowledgements** The authors are grateful to the local authority public health teams for providing pseudonymised National Child Measurement Programme data, school nurses and National Child Measurement Programme data collection teams and National Child Measurement Programme participants. They are grateful to Professor Mario Cortina-Borja (University College London Great Ormond Street Institute of Child Health) for supporting the development of an algorithm to randomly assign a day of measurement. They are also grateful to the data controllers of the Discovery Programme and to the GPs and their practice teams for allowing the use of their patient records, to the Clinical Effectiveness Group for providing access to their curated high-quality dataset and to the population in east London from whom the data are derived. This work uses data provided by patients and collected by the NHS as part of their care and support.

**Contributors** CD obtained funding for the study. NF, CD and GH conceptualised and designed the analyses. NF, MW, MM, GH and CD contributed to the development of the methodology. NF carried out the literature search, conducted the analyses, generated tables and figures and drafted the initial manuscript. NF, MW, MM, GH, LG and CD contributed to the interpretation of analyses and reviewed and revised the manuscript.NF, MW, MM, GH, LG and CD were involved in writing the paper and had final approval of the submitted and published manuscript. NF is the guarantor and accepts full responsibility for the conduct of the study, had access to the data, and controlled the decision to publish. The corresponding author attests that all listed authors meet authorship criteria and that no others meeting the criteria have been omitted.

**Funding** This work was supported by ADR UK (Administrative Data Research UK), an Economic and Social Research Council investment (part of UK Research and Innovation) (grant number: ES/X00046X/1). This research was also supported by grants from Barts Charity (ref: MGU0419 and MGU0504). This work was supported by the UK Prevention Research Partnership (MR/S037527/1), which is funded by the British Heart Foundation, Cancer Research UK, Chief Scientist Office of the Scottish Government Health and Social Care Directorates, Engineering and Physical Sciences Research Council, Economic and Social Research Council, Health and Social Care Research and Development Division (Welsh Government), Medical Research Council, National Institute for Health Research, Natural Environment Research Council, Public Health Agency (Northern Ireland), The Health Foundation and Wellcome.

**Competing interests** None declared.

**Patient and public involvement** Patients and/or the public were not involved in the design, or conduct, or reporting, or dissemination plans of this research.

**Patient consent for publication** Not applicable.

**Ethics approval** The analyses of linked pseudonymised school measurement programme and GP data were approved by the respective data controllers under data processing agreements which allow linkage of pseudonymised school measurement programme data between the research organisation and each local authority public health team. This study is a secondary analysis of de-identifiable data and no further ethics approval was required.

**Provenance and peer review** Not commissioned; externally peer reviewed.

**Data availability statement** Data may be obtained from a third party and are not publicly available. Access to primary care data is enabled by data sharing agreements between the Discovery Data Service and the data controllers. The Discovery Programme Board has approved data access by the REAL Child Health Programme team for research on the condition that it is not onwardly shared. National Child Measurement Programme data were accessed under data processing agreements with each of the local authorities as data controllers in line with Public Health England guidance. These agreements preclude onward sharing of data.

**ORCID iDs**
Nicola Firman http://orcid.org/0000-0001-5213-5044
Lucy Griffiths http://orcid.org/0000-0001-9230-624X

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
