## [Reviewer comments · BMJ Paediatrics Open]

ARTICLE DETAILS

TITLE (PROVISIONAL)	Is obesity more likely among children sharing a household with an older child with obesity? Cross-sectional study of linked National Child Measurement Programme data and electronic health records
AUTHORS	Firman, Nicola Wilk, Marta Marszalek, Milena Griffiths, Lucy Harper, Gill Dezateux, Carol

VERSION 1 – REVIEW

REVIEWER	Dr. Chester Kalinda University of Global Health Equity, Institute of Global Health
REVIEW RETURNED	08-Feb-2024

GENERAL COMMENTS	1. The study outcome is the obesity status of the index child. if the reference child (child 2) lives in the same environment as the index child, then the health outcome of one child is potentially likely to be serially correlated with that of the index child. Did the authors check for serial correlation, and how did they deal with this? 2. In the data analysis, there is this statement: "We conducted linear regression to estimate the effect of a one-unit increase in the oldest child's BMI z-score on the index child's BMI z-score." Often times, the Body Mass Index (BMI) z-score is not or may not be normally distributed on its own, and thus diagnostic tests to establish its distribution are important. How did the authors deal with this? Normality tests? 3. From Table S4, which has been exceptionally done. I notice that the community-level factor "local authority" has been modelled together with household levels. Would it have been better if this were modelled differently considering that some factors that are social service-related or health care-related—access, quality, and distance to health facilities—and their provision may vary from one community to another, thus impacting the child's outcome?
--

REVIEWER	Dr. Staffan Mårild University of Gothenburg Institute of Clinical Sciences
REVIEW RETURNED	29-Feb-2024

GENERAL COMMENTS	Review of manuscript ID bmjpo-2024-002533: "is obesity more likely among children sharing a household with an older child with obesity?"
---

	I find the paper interesting with very impressive and advanced computer- and register work identifying siblings, both living with obesity, in a large population of children. The data were pseudonymised and several different data-sources were used and linked in the process to answer the research question. My main concern is the implications of the study for clinicians, are the findings after all the cumbersome computer-work useful at all? See comments on the discussion below Abstract: In order to make the paper easier to follow for international readers, the ages of both the younger and the older children should be given as well as the BMI-z-scores used as cut-offs for the different weight-categories. The thoughts about further research is questionable to include in the conclusion even if it is a good idea. Abbreviations: There are a great number of abbreviations, NCMP, NEL, HER, DDS, GP etc. a list of all abbreviations would be very helpful, and I also suggest you give a nick-name for each one of them, eg. NCMP, "measurement data" or similar; UPRN, "property data" etc . Key messages:  - second sentence, "Less is known..." : This was not possible to address in the paper and seems to be a somewhat irrelevant statement. - regarding the last paragraph under "How this study .. ": -- why does the household approach "encompass a broader range of factors..." ? any kind of intervention on 4-5 och 10-11 year-old children with obesity must be family and environment oriented. (see also the discussion below) Introduction: the background describes the strong impact of socio-economic and environmental factors on the development of obesity. The genetic aspects must be mentioned in addition, see comments below re the discussion. The introduction gives a background to the study objectives, but there no research questions or hypothesis are presented. Methods: the above suggestion of a list explaining the different abbreviations and nicknames would help a lot for readers here. Here some mote specific questions related to methods:  - the children/ families were invited to the measurement program and the children in the study solely attended state-maintained schools. This indicates that there is some selection bias. The census data for 4-5- and 10–11-year-old children in the four areas in north-east London must be able to use; even if the study population per se is very large, the reader does not know the number of the total populations. - The day of measurement was randomly assigned but, what about seasonal variations in height? Childrens height-growth is seasonal, since they grow mainly in the summer. - Height and weight measurements were used to calculate BMI: no statistics is given for these measures (neither for the BMI), e.g. height-measures are crucial in calculation of BMI and for its distribution. Height is often problematic to measure and varies greatly by measuring staff. Were there no height measures the had to be excluded? - Clinical data were obtained from the GP-records. Were there no exclusions of individuals due to severe diseases or medical problems?
--	---

	- The whole study is in the virtual world, which is a strength in many ways, but it also gives you an artificial feeling. Was some kind of validation of the main findings done or was any sensitivity analysis performed? - “Outcome of interest” (page 6) : the BMI z-score is not to be found in the paper ? - Explanatory variables: for the classification of the four weight categories the UK1990 reference standard was used. The application in GB of these standards is of course natural. For a more general and international public it would be valuable to compare these data with the IOTF (T Cole) standards and also add cut-offs as BMI z-scores, at least in a separate supplementary table but also in the text if possible. The British centiles are very abstract and hard to comprehend and giving BMI-z-scores in addition would be valuable. The different weight categories do not correspond to the IOTF, the underweight group is under 2 c (very extreme), the healthy weight is very broadly defined from 2nd to the 91 centile. This is probably according to reference 19 in the paper, but might be of interest to comment in the discussion? - The IMD classification should be explained, it is related to neighbourhood characteristics, but how? Results: The results are well written and presented in text and tables, however, legends to tables and figures are hard to find, perhaps something that depends on this pre-print? Those excluded because of non-residential households (n=3903) seem important to characterise and give more information about: do they live in the street? Do their BMI differ much from the majority? Discussion: the authors seem to have followed the guidelines of the journal 100%. In my view the discussion is extremely meagre and incomplete, perhaps due to the attempt to stay short? It seems very important here to expand the discussion somewhat and give a better support to the conclusion. The strengths and weaknesses of the study covers the larger part of the discussion, too large in my view. One weakness is that the sibling’s biological relation was not possible to assess, I find a bit annoying that the authors try to make this weakness a strength by stating: “this is the first time that other children in the household..... etc”; I am quite sure this is not true (“first time”) and I suggest this text is changed. The last sentence in this section (bottom page 10) is a more general comment and is better to put in the paragraph I suggest below. Weakness in the design: The study was performed in four areas with quite deprived populations. This is really a great weakness in the study: differences in socio-economic and neighbourhood characteristics of the populations would yield important information on the sibling-obesity phenomenon. strengths and weaknesses in relation to other studies is suggested in the guidelines; I suggest that the discussion is changed in violation to the guidelines in order to make it broader and cover many other important issues:  - environment: one important aspect here are related to the built environment, housing quality, green areas, security, transportation, “food deserts”; - another relates to family environment and socio-economic situation (education, income, employment) - genetics, it is important to also mention the strong (poly-)genetic impact on the development of obesity, the “thrifty genotype” and
--	--

	the gen-environment interaction behind the obesity epidemic; most likely there is a genetic contribution to the presence of two children with obesity in a family, besides the environmental impact. All these aspects are important to discuss shortly with relevant literature references. The IOTF comments mentioned above are also important to discuss shortly implications for clinicians: If you meet a 4-5 year-old child with obesity in the clinic, there is hope-fully some program for intervention. Usually these programs include “a life-style” interventions among parents, all family members, the extended family (grand-parents), school etc. Most likely the health-worker will learn about any sibling with obesity in such a program. The study findings point out the great likelihood of obesity in siblings. On the other hand: there is a great risk that families with 2 children with obesity feel stigmatised and might even refuse clinical intervention. implications for policymakers: The problem with stigmatisation is actually great. In addition, the rate of drop-outs from interventions are substantial. Therefore health promotion programs are very important and actually interventions having some effect. For English settings, the book by Henry Dimbleby “Ravenous” is recommended. unanswered questions and future research: A similar study with less deprived population and in other parts o England would be valuable for comparison. The suggestion of qualitative research in abstract conclusion is interesting. To sum up: The study have some weaknesses in design and the outcome is of relative little value in the clinical work
--	--

VERSION 1 – AUTHOR RESPONSE

Reviewer 1		
1	The study outcome is the obesity status of the index child. if the reference child (child 2) lives in the same environment as the index child, then the health outcome of one child is potentially likely to be serially correlated with that of the index child. Did the authors check for serial correlation, and how did they deal with this?	Thank you for this thoughtful comment. We did not consider serial correlation in our data as we only had a single BMI measurement from the school measurement programme. We have previously shown that GP records of BMI are incomplete and biased, which we why we rely upon the school measurement programme for data on BMI/weight status (doi: 10.1111/ijpo.12772)

2	In the data analysis, there is this statement: "We conducted linear regression to estimate the effect of a one-unit increase in the oldest child's BMI z-score on the index child's BMI z-score." Often times, the Body Mass Index (BMI) z-score is not or may not be normally distributed on its own, and thus diagnostic tests to establish its distribution are important. How did the authors deal with this? Normality tests?	Thank you for highlighting this. We have checked and the youngest and oldest children's BMI z-scores are normally distributed. Consequently, the regression residuals are sampled from a normal distribution. We have added this information to the statistical analyses section.
3	From Table S4, which has been exceptionally done. I notice that the community-level factor "local authority" has been modelled together with household levels. Would it have been better if this were modelled differently considering that some factors that are social service-related or health care-related—access, quality, and distance to health facilities—and their provision may vary from one community to another, thus impacting the child's outcome?	Thank you for this comment. We employed a staggered, forward and backward selection procedures to reach our final multivariable models. We first introduced all individual-level demographic characteristics, followed by variables relating to the older child's measurement (sex concordance and time difference between the two measurements). Next, we introduced all household-level variables, and finally we introduced area-level variables (deprivation and local authority). We included area-level variables as well as individual factors although neither were retained in the final stratified models given the correlation between local authority, deprivation and ethnic background in north-east London.
Reviewer 2		
1	Abstract: In order to make the paper easier to follow for international readers, the ages of both the younger and the older children should be given as well as the BMI-z-scores used as cut-offs for the different weight-categories. The thoughts about further research is questionable to include in the conclusion even if it is a good idea	Thank you for this comment. We have added the age range and median age of the study participants to the

		abstract. The UK1990 cut-offs for defining childhood obesity are well documented and published in many international journals.
2	Abbreviations: There are a great number of abbreviations, NCMP, NEL, HER, DDS, GP etc. a list of all abbreviations would be very helpful, and I also suggest you give a nick-name for each one of them, eg. NCMP, “measurement data” or similar; UPRN, “property data” etc.	Thank you for highlighting this. We have reviewed the manuscript and reduced the use of abbreviations throughout.
3	Key messages: second sentence, “Less is known....” : This was not possible to address in the paper and seems to be a somewhat irrelevant statement	Thank you for this suggestion, we have updated the key message.
4	Key messages: regarding the last paragraph under “How this study .. “: -- why does the household approach “encompass a broader range of factors....” ? any kind of intervention on 4-5 och 10-11 year-old children with obesity must be family and environment oriented (see also the discussion below)	Thank you for this comment, we have updated this section.
5	Introduction: the background describes the strong impact of socio-economic and environmental factors on the development of obesity. The genetic aspects must be mentioned in addition, see comments below re the discussion. The introduction gives a background to the study objectives, but there no research questions or hypothesis are presented	Thank you for this suggestion. We have updated the introduction to acknowledge a wider range of factors contributing to childhood obesity, including the genetic aspects. We have also added our hypothesis.
6	Methods: the children/ families were invited to the measurement program and the children in the study solely attended state-maintained schools. This indicates that there is some selection bias. The census data for 4-5- and 10–11-year-old children in the four areas in north-east London must be able to use; even if the study population per se is very large, the reader does not know the number of the total populations	Thank you for this comment. We have added more information about participation in the NCMP in the study population section and acknowledged the implications of this in the limitations section. We are unable to compare to Census estimates as these are only published in five-year age bands.
7	Methods: the day of measurement was randomly assigned but, what about seasonal variations in height? Childrens height-growth is seasonal, since they grow mainly in the summer	Thank you for this question. We received observed height measurements which were not estimated or imputed. We received

		the month and year of NCMP measurement and randomly assigned the day within the recorded month and year.
8	Methods: height and weight measurements were used to calculate BMI: no statistics is given for these measures (neither for the BMI), e.g. height-measures are crucial in calculation of BMI and for its distribution. Height is often problematic to measure and varies greatly by measuring staff. Were there no height measures the had to be excluded?	Thank you for this question. School nursing teams who collect the NCMP data are trained to use standardised protocols to measure height and weight. The NCMP employs quality assurance checks according to validation guidance (https://digital.nhs.uk/services/national-child-measurement-programme/it-system/validation-of-national-child-measurement-programme-data). We have only received NCMP records that have been through these validation checks and have added this detail to the methods section.
9	Methods: clinical data were obtained from the GP-records. Were there no exclusions of individuals due to severe diseases or medical problems?	Thank you for this question. The NCMP excludes children who cannot stand unaided or in whom accurate results cannot be taken due to conditions such as cerebral palsy, a prosthetic leg, or a growth disorder.
10	Methods: the whole study is in the virtual world, which is a strength in many ways, but it also gives you an artificial feeling. Was some kind of validation of the main findings done or was any sensitivity analysis performed?	Thank you for this question. This study uses routinely collected, high quality, real world data which has not been primarily collected for research

		so we are unable to confirm the accuracy of individual measurements, however at a population level numerous studies have shown these to be reliable. We did not undertake sensitivity analyses.
11	Methods: "Outcome of interest" (page 6) : the BMI z-score is not to be found in the paper ?	We report results of linear regression estimating the impact of a one unit increase in the older child's BMI z-score on the younger child's BMI z-score (page 9).
12	Methods: explanatory variables: for the classification of the four weight categories the UK1990 reference standard was used. The application in GB of these standards is of course natural. For a more general and international public it would be valuable to compare these data with the IOTF (T Cole) standards and also add cut-offs as BMI z-scores, at least in a separate supplementary table but also in the text if possible. The British centiles are very abstract and hard to comprehend and giving BMI-z-scores in addition would be valuable. The different weight categories do not correspond to the IOTF, the underweight group is under 2 c (very extreme), the healthy weight is very broadly defined from 2nd to the 91 centile. This is probably according to reference 19 in the paper, but might be of interest to comment in the discussion?	Thank you for this comment. In the UK it is recommended to use the UK1990 cut-offs. We have added a new supplementary table (Table S2) showing the weight status distribution among index children using IOTF cut-offs, as well as the UK1990 clinical cut-offs.
13	Methods: the IMD classification should be explained, it is related to neighbourhood characteristics, but how?	Thank you for this suggestion. We have added further information about the derivation of the Index of Multiple Deprivation measure in the Table 1 footnote.
14	Results: the results are well written and presented in text and tables, however, legends to tables and figures are hard to find, perhaps something that depends on this pre-print?	We agree it is difficult to identify figures with their titles and legends in the pre-print PDF format!
15	Results: those excluded because of non-residential households (n=3903) seem important to characterise and give more information about: do they live in the street? Do their BMI differ much from the majority?	Thank you for this suggestion. We have added the information about the

		characteristics of these children to the Figure 1 footnote.
16	Discussion: the authors seem to have followed the guidelines of the journal 100%. In my view the discussion is extremely meagre and incomplete, perhaps due to the attempt to stay short? It seems very important here to expand the discussion somewhat and give a better support to the conclusion.	Thank you for these thoughtful comments about the discussion in general. We have updated the discussion, shortening the strengths and limitations section and adding to the comparison with existing literature section in line with these suggestions.
17	The strengths and weaknesses of the study covers the larger part of the discussion, too large in my view. One weakness is that the sibling's biological relation was not possible to assess, I find a bit annoying that the authors try to make this weakness a strength by stating: "this is the first time that other children in the household..... etc"; I am quite sure this is not true ("first time") and I suggest this text is changed. The last sentence in this section (bottom page 10) is a more general comment and is better to put in the paragraph I suggest below	Thank you for this suggestion. We have updated the strengths and limitations section to reflect this comment.
18	Weakness in the design: The study was performed in four areas with quite deprived populations. This is really a great weakness in the study: differences in socio-economic and neighbourhood characteristics of the populations would yield important information on the sibling-obesity phenomenon.	Thank you for this comment. We have added this to the limitations.
19	Strengths and weaknesses in relation to other studies is suggested in the guidelines; I suggest that the discussion is changed in violation to the guidelines in order to make it broader and cover many other important issues: - environment: one important aspect here are related to the built environment, housing quality, green areas, security, transportation, "food deserts"; - another relates to family environment and socio-economic situation (education, income, employment) - genetics, it is important to also mention the strong (poly-)genetic impact on the development of obesity, the "thrifty genotype" and the gen-environment interaction behind the obesity epidemic; most likely there is a genetic contribution to the presence of two children with obesity in a family, besides the environmental impact. All these aspects are important to discuss shortly with relevant literature references	Thank you for these suggestions. We have updated the comparison with existing literature section to include mention of these issues to acknowledge the complicated associations and interactions between obesity and genetic factors, social and environmental factors.
20	The IOTF comments mentioned above are also important to discuss shortly	Thank you for this suggestion. In the UK it is recommended to use the UK1990 cut-offs. A recognised limitation of these cut-offs is that international comparisons are not possible. We have added this to the limitations and reported

		estimates using International Obesity Task Force cut-offs in supplementary Table S2.
21	Implications for clinicians: If you meet a 4-5 year-old child with obesity in the clinic, there is hope-fully some program for intervention. Usually these programs include “a life-style” interventions among parents, all family members, the extended family (grand-parents), school etc. Most likely the health-worker will learn about any sibling with obesity in such a program. The study findings point out the great likelihood of obesity in siblings. On the other hand: there is a great risk that families with 2 children with obesity feel stigmatised and might even refuse clinical intervention	Thank you for this comment. Parents of children participating in the NCMP are sent a letter informing them of their child’s weight status and advising them to speak with their GP if they are concerned about their child’s weight. In north-east London, there are few interventions available for children with obesity. A GP will only find out if there are other children in the family living with obesity if the family consult the GP. A household approach may offer some way of prioritising services given our finding that there is increased likelihood of obesity in children sharing the same household. We do not have data to indicate whether or not children are biologically related. This is important given that the prevalence of blended and non-nuclear family structures is increasing.
22	Implications for policymakers: The problem with stigmatisation is actually great. In addition, the rate of drop-outs from interventions are substantial. Therefore health promotion programs are very important and actually interventions having some effect. For English settings, the book by Henry Dimbleby “Ravenous” is recommended	Thank you for this thoughtful comment, we agree with the reviewer.
23	Unanswered questions and future research: A similar study with less deprived population and in other parts o England would be valuable for comparison	Thank you for this suggestion. We have added this to the implications section.
24	The suggestion of qualitative research in abstract conclusion is interesting.	Thank you for this comment.